

# Epithelial, metabolic and innate immunity transcriptomic signatures differentiating the rumen from other sheep and mammalian gastrointestinal tract tissues

Ruidong Xiang[1], Victor Hutton Oddy[2], Alan L. Archibald[3], Phillip E. Vercoe[4] and Brian P. Dalrymple[1]

[1] CSIRO Agriculture, St. Lucia, QLD, Australia

[2] NSW Department of Primary Industries, Beef Industry Centre, University of New England, Armidale, NSW, Australia

[3] The Roslin Institute and Royal (Dick) School of Veterinary Studies, University of Edinburgh, Easter Bush, UK

[4] School of Animal Biology and Institute of Agriculture, The University of Western Australia, Perth, Western Australia, Australia

Corresponding author
Brian P. Dalrymple,
brian.dalrymple@csiro.au

## ABSTRACT

**Background.** Ruminants are successful herbivorous mammals, in part due to their specialized forestomachs, the rumen complex, which facilitates the conversion of feed to soluble nutrients by micro-organisms. Is the rumen complex a modified stomach expressing new epithelial (cornification) and metabolic programs, or a specialised stratified epithelium that has acquired new metabolic activities, potentially similar to those of the colon? How has the presence of the rumen affected other sections of the gastrointestinal tract (GIT) of ruminants compared to non-ruminants?

**Methods.** Transcriptome data from 11 tissues covering the sheep GIT, two stratified epithelial and two control tissues, was analysed using principal components to cluster tissues based on gene expression profile similarity. Expression profiles of genes along the sheep GIT were used to generate a network to identify genes enriched for expression in different compartments of the GIT. The data from sheep was compared to similar data sets from two non-ruminants, pigs (closely related) and humans (more distantly related).

**Results.** The rumen transcriptome clustered with the skin and tonsil, but not the GIT transcriptomes, driven by genes from the epidermal differentiation complex, and genes encoding stratified epithelium keratins and innate immunity proteins. By analysing all of the gene expression profiles across tissues together 16 major clusters were identified. The strongest of these, and consistent with the high turnover rate of the GIT, showed a marked enrichment of cell cycle process genes ($P = 1.4E-46$), across the whole GIT, relative to liver and muscle, with highest expression in the caecum followed by colon and rumen. The expression patterns of several membrane transporters (chloride, zinc, nucleosides, amino acids, fatty acids, cholesterol and bile acids) along the GIT was very similar in sheep, pig and humans. In contrast, short chain fatty acid uptake and metabolism appeared to be different between the species and different between the rumen and colon in sheep. The importance of nitrogen and iodine recycling in sheep

was highlighted by the highly preferential expression of *SLC14A1*-urea (rumen), RHBG-ammonia (intestines) and *SLC5A5*-iodine (abomasum). The gene encoding a poorly characterized member of the maltase-glucoamylase family (MGAM2), predicted to play a role in the degradation of starch or glycogen, was highly expressed in the small and large intestines.

**Discussion.** The rumen appears to be a specialised stratified cornified epithelium, probably derived from the oesophagus, which has gained some liver-like and other specialized metabolic functions, but probably not by expression of pre-existing colon metabolic programs. Changes in gene transcription downstream of the rumen also appear have occurred as a consequence of the evolution of the rumen and its effect on nutrient composition flowing down the GIT.

## INTRODUCTION

The ruminants, of which sheep, cattle, buffalo and goats are the major domesticated species, are now the most numerous large herbivores on earth. Their success is partly due to their specialized forestomachs, the rumen complex (the rumen, reticulum and omasum), and to rumination, the process of recycling the partially digested material via the mouth to reduce particle size and increase rate of fermentation (*Hofmann, 1989*). The forestomachs follow the oesophagus and precede the abomasum (the equivalent of the stomach of non-ruminants) (*Hofmann, 1989*). The evolutionary origin of the rumen is the subject of debate with out-pouching of the oesophagus, or of the stomach, as the most likely origins (*Beck, Jiang & Zhang, 2009*; *Langer, 1988*). The primary chambers of the rumen facilitate the action of a complex mixture of micro-organisms to ferment a portion of the plant polysaccharides (including starch, xylan and cellulose) and lipids to short chain volatile fatty acids (SCFAs), principally acetate, butyrate and propionate (*Bergman, 1990*). The SCFAs are the primary energy source in carbon of ruminants, and the rumen is the major site of their uptake.

From the rumen, partially processed plant material, nutrients, and micro-organisms pass through the omasum and enter the conventional gastrointestinal system: the abomasum, and the small and large intestines for further digestion and fermentation (in the large intestine). The abomasum is primarily a digestive organ lowering the pH of the rumen fluid and facilitating the first step of proteolysis prior to more extensive degradation in the duodenum and absorption of amino acids and small peptides. Pancreatic RNAses degrade microbial RNA in the small intestine contributing to nitrogen availability. On pasture, roughage or grass diets only small amounts of starch escape fermentation in the rumen and the remaining starch is generally digested in the small intestine, providing limited amounts of glucose (*Deckardt, Khol-Parisini & Zebeli, 2013*). Depending on the dietary source larger amounts of starch may escape fermentation in the rumen (*Huntington,*

*1997*). As a consequence glucose is not a major source of carbon in ruminants, and the liver is not a major site of (fatty acids) FA synthesis (*Ingle, Bauman & Garrigus, 1972*). Biohydrogenation processes in the rumen (*Van Nevel & Demeyer, 1996*) increase the saturation of fatty acids (*Jenkins et al., 2008*; *Van Nevel & Demeyer, 1996*), and lipids that escape fermentation in the rumen are taken up in the small intestine. Fermentation of the remaining carbohydrates, lipid etc. occurs in the large intestine/hindgut. The hindgut is responsible for 5–10% of the total digestion of carbohydrates (*Gressley, Hall & Armentano, 2011*) and for 8–17% of total production of SCFAs (*Hoover, 1978*). This contribution of hindgut fermentation may be altered on high grain diets (*Fox et al., 2007*; *Mbanzamihigo, Van Nevel & Demeyer, 1996*). The overlap in functions of the rumen and the hindgut raises the question of whether the equivalent processes in the two tissues are undertaken by the same proteins and pathways; that is co-option of the hindgut program by the rumen, or by different proteins and pathways resulting from convergent evolution.

Unlike the stomach and subsequent segments of the GIT the rumen surface is a stratified squamous epithelium that is cornified and keratinized to protect the rumen from physical damage from the ingested plant material (*Scocco et al., 2013*). Due to the large numbers of microorganisms in the rumen it is also exposed to colonization of surfaces and potential attack from these organisms. The nature of the defences and the interaction between the surface of the rumen and the microbial populations has not been investigated in detail.

Herein, we utilised the latest sheep genome and transcriptome data (*Jiang et al., 2014*) to further dissect gene expression features of the ruminant GIT. We analyze the transcriptomes of six GIT tissue/cell types covering the majority of the sheep GIT in the context of reference samples from two other tissues with stratified squamous epithelium (skin and tonsil), another component of the immune system (spleen), and two non-epithelial tissues (liver and muscle). Further, we systematically compared our results with existing transcriptome data from the human and pig gastrointestinal tracts and with relevant literature using candidate gene/protein based approaches. Our major aims were to identify: (i) the distinctive features of ruminant GIT, (ii) the common features shared between ruminant and mammalian GIT and (iii) the developmental origin of the rumen.

## METHODS

### Data acquisition and statistical analysis

No new primary datasets were generated in this work, the major secondary datasets are included in the supplementary material. The sample preparation procedures and sequencing of the RNA are described in *Jiang et al. (2014)* and experimental animal information is specified in Table S1. Briefly, tissue samples were obtained from a trio of Texel sheep, i.e., ram ($r$), ewe ($e$) and their lamb ($l$). RNA was prepared and sequenced using stranded Illumina RNA-Seq with a yield of 70–150 million reads per tissue sample. 26 files of RNA sequence alignment data in the BAM format for 11 tissue/cell types, including skin ($n = 3$), tonsil ($n = 1r$), ventral rumen ($n = 3$), abomasum ($n = 3$), duodenum ($n = 1r$), caecum ($n = 2$, $r$ and $l$), colon ($n = 3$), rectum ($n = 3$), spleen ($n = 2$, $r$ and $l$), liver ($n = 2$, $r$ and $e$) and muscle ($n = 3$), were downloaded from the Ensembl sheep

RNA sequencing archive, Oar_v3.1 (*Huttenhower et al., 2009*; *Jiang et al., 2014*). Detailed animal and gender distribution can be found in Fig. S1. Detailed raw RNA sequencing data from the same samples was also retrieved from the European Nucleotide Archive (ENA), study accession PRJEB6169. The raw mapping counts for each gene were calculated from the downloaded BAM files and the Ensembl sheep gene models (Sheep Genome v3.1, http://www.ensembl.org/Ovis_aries/Info/Index), with additional gene models for genes at the epidermal differentiation complex (EDC) locus not included in the Ensembl sheep gene models (*Jiang et al., 2014*), using HTSeq in the Python environment (*Anders, Pyl & Huber, 2015*). The raw count data was normalized and clustered with DEseq2 (*Love, Huber & Anders, 2014*) to produce PCA plots and variance-stabilizing transformed gene expression values for network analysis described below. DEseq2 produced PCA sample clustering was further tested for significance using a $k$-means method and bivariate $t$-distributions based on the eigenvalues of the principle components. Calculation was performed using the stat_ellipse package (*2012*) and the raw outputs were presented in ggplot2 in R. EdgeR (*Robinson, McCarthy & Smyth, 2010*) in Bioconductor in R v3.1.3 was used to analyse gene differential expression. After filtering for transcripts with at least 1 count per million in at least one of the 11 tissues, data was analysed using the Analysis of Variance-like procedure (special feature in EdgeR) and fitted to a simple model: $y = tissue_i + animal_j + e_{ij}$ where $y$ is raw transcript counts, $tissue_i$ ($i = 11$) is 11 types of tissues and $animal_j$ ($j = 3$) is the adjustment of types of animal (lamb, ram and ewe). Transcripts with significance levels ($P$) < 0.01 and false discovery rate (FDR) < 0.01 for tissue effects and differentially expressed in at least one of the 11 tissues were identified.

## Co-expression network analysis

Variance-stabilizing transformed RNA sequencing expression values have properties similar to normalized microarray expression values in terms of network analysis (*Giorgi, Del Fabbro & Licausi, 2013*) and raw counts of differentially expressed (FDR < 0.01) transcripts were variance-stabilizing transformed (*Durbin et al., 2002*) using DEseq2. Transformed expression values were analyzed for co-expression using PCIT (*Hudson, Dalrymple & Reverter, 2012*; *Reverter & Chan, 2008*) in R v3.1.3 (*Watson-Haigh, Kadarmideen & Reverter, 2010*). To reduce the complexity of the network the PCIT output was filtered for pairs of genes with a correlation coefficient >0.9 and visualized in Cytoscape v3.1.2 (*Shannon et al., 2003*). The network cluster algorithm 'community cluster' within the GLay plugin (*Su et al., 2010*) of Cytoscape was used to subdivide the large network and identify explanatory sub-networks in an iterative manner until no obvious sub-network was observed in the large network. Pig genes assigned to 10 clusters showing differential expression in the pig GIT (*Freeman et al., 2012*) were mapped to sheep genes based on their gene symbols. The probability of over or under representation of pig GIT genes in a sheep GIT gene cluster was calculated using the hypergeometric distribution (*Andrews, Askey & Roy, 1999*). Functional enrichment of shared sets of genes within sheep clusters was analyzed using GOrilla (*Eden et al., 2009*) to identify biological pathways.

## Gene expression pattern clustering

The transcripts present in the gene networks described above, and with an ANOVA $P < 0.01$ and a FDR $< 0.01$, were included in $k$-mean clustering in R v3.1.3 based on $\log_2$ fold change across 11 tissues with abomasum being the reference. The $k$-mean analysis aimed to identify expression patterns to represent transcript groups showing elevated expression levels for the following sets of tissues v. the remaining tissues: (1) all GIT tissues, i.e., rumen, abomasum, duodenum, caecum, colon and rectum, (2) rumen and abomasum, (3) rumen and intestinal tissues, (4) abomasum and intestinal tissues, (5) rumen, (6) abomasum, (7) intestinal tissues, (8) rumen and skin, (9) rumen and tonsil, (10) rumen, skin and tonsil, (11) spleen, duodenum, caecum, and colon. The transcript names are determined based on the tissue(s) where included transcripts showed the highest expression. We filtered the identified transcript clusters with the criteria that (1) the average absolute expression of the transcript at the highest expressed tissue $> 3$ counts per million, (2) the $\log_2$ expression fold difference of expression of the transcript from the tissue within the reference tissue group with the highest expression to the tissue within the elevated expressed tissue group with the highest expression, be $>0.5$, and (3) from the tissue with the highest expression to the tissue with the lowest expression within the elevated tissue group be $<0.5$. The final expression of each transcript is presented in the format of log2 Fragments Per Kilobase of exon per Million fragments mapped (FPKM). Selected gene members and associated pathways were presented in heat maps based on their log2 FKPM values using GENE-E.

To understand the GIT associated SLC family genes, we performed a network analysis of expression as above. The PCIT output of network matrix was filtered for correlation coefficient $> 0.7$, clustered by GLay (*Su et al., 2010*) and visualized in Cytoscape v3.1.2 (*Shannon et al., 2003*).

## Comprehensive transcript annotation

To complement the sheep genome annotation, we used multiple annotation sources and software to identify the function of the products encoded by the identified transcripts. Firstly, the transcripts of interest, both with and without a gene symbol, were validated for existence in the sheep genome, using comparisons of the sheep gene within the locus with its ortholog(s) in human and cattle from Ensembl and NCBI. Secondly, GO was used to annotate genes. Thirdly, the functions and annotations of the genes were searched in Ensembl and NCBI, if no available description or gene information were identified, the biomedical literature was searched with GenCLiP 2.0 (*Wang et al., 2014*). When multiple biomedical functions were listed, functions related to gastrointestinal activity were prioritized for annotation. Fourthly, for a subset of genes Unigene (*McGrath, Bolling & Jonkman, 2010*), Genevestigator (*Hruz et al., 2008*) and GeneAtlas (*Frezal, 1998*) were used to identify transcript expression patterns in cattle and humans respectively. Protein sequences analysis was performed using Radar (*Heger & Holm, 2000*), to identify amino acid sequence repeats, and NetOGlyc 4.0 (*Steentoft et al., 2013*), to identify glycosylation sites.

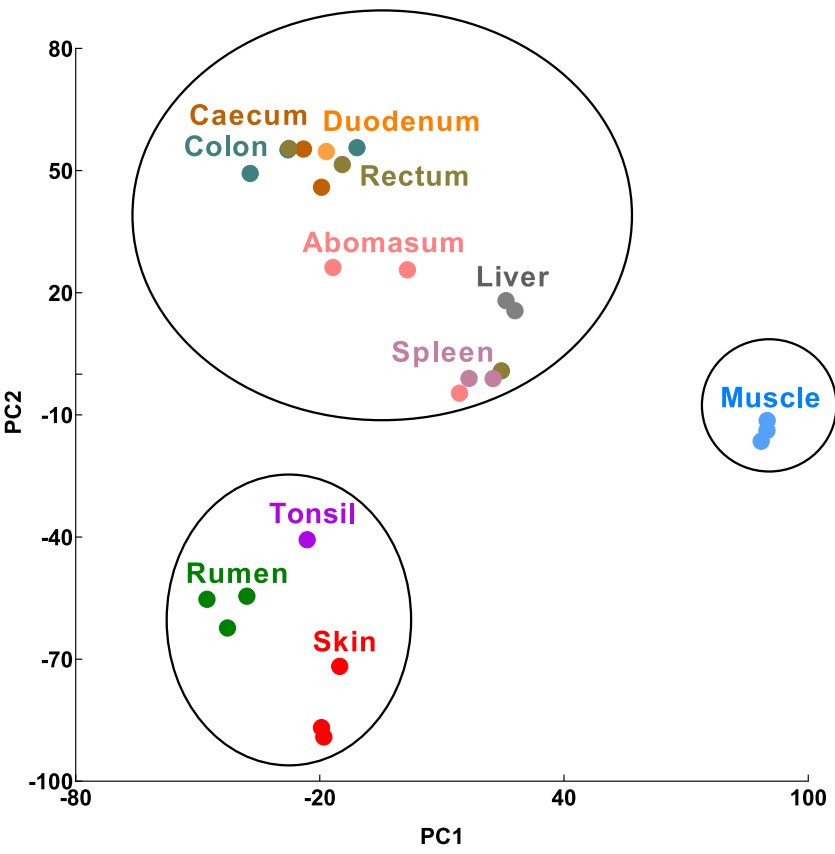

**Figure 1** **Transcriptomic sample clustering.** Each dot represents one tissue sample from a single animal. Circles indicate significant clusters (confidence interval = 95%). Raw PCA plots are available (Figure S1).

## Data access

No new primary datasets were generated in this work, the major secondary datasets are included in the Supplemental Information.

## RESULTS AND DISCUSSION

### Clustering of sheep GIT tissue transcriptomes

We performed principal component analysis (PCA) using RNA-Seq data from six GIT (rumen, abomasum, duodenum, caecum, colon and rectum), two epithelial (skin and tonsil), an immune (spleen) and two reference (liver and muscle) tissue/cell types from a trio of Texel sheep (ram, ewe and lamb (*Jiang et al., 2014*)). We included a total of 26 tissue samples, a similar tissue sample coverage to a previous transcriptomic study of the pig GIT (*Freeman et al., 2012*) to which the results of this analysis will be compared below. Three clusters of tissues were identified at the 95% confidence interval: cluster 1, skin, tonsil and rumen, cluster 2, muscle, and cluster 3, liver, spleen and the remaining GIT tissues (Fig. 1A, Fig. S1A).

**Table 1  Gene Ontology enrichments of clusters.**

| Cluster | GO-term | FDR corrected $P$-value[a] |
|---|---|---|
| Rumen | EDC locus[b] | 7.1E-13[c] |
| Epithelia-rumen-tonsil | EDC locus[b] | 8.6E-15[c] |
|  | Defense response to fungus | 8.6E-03 |
| Epithelia-rumen bias | Keratinization | 2.4E-04 |
| Epithelia-all | – | – |
| Epithelia-large intestine | Desmosome organization | 4.7E-03 |
| Epithelia-GI-liver | Cell junction organization | 6.3E-03 |
| Abomasum-intestine | – | – |
| Intestine-low in rectum | – | – |
| Large intestine | Regulation of chloride transport | 4.5E-05 |
| Intestine | – | – |
| Epithelia-intestine | Cell cycle process | 1.4E-46 |
| Abomasum | Digestion | 3.8E-02 |
| Small intestine | – | – |
| Rumen-abomasum | Platelet aggregation | 2.2E–04 |
| Rumen-intestine-liver | Flavonoid biosynthetic process | 5.5E–10 |
| Intestine-spleen | Humoral immune response | 4.5E–02 |

**Notes.**
[a]Top significantly enriched pathway selected from GOrilla analysis (see 'Methods') for each input gene cluster.
[b]Genes in the EDC locus of the sheep genome.
[c]Enrichment of EDC locus genes was calculated using the hypergeometric distribution.

## Identification of common and specific GIT and epithelial transcriptomic signatures

To identify the genes driving the clustering of the tissues we identified those transcripts with an ANOVA $P < 0.01$ and a false discovery rate (FDR) $< 0.01$, for differential expression in at least one tissue versus the other tissue types. This multi-tissue comparison reduced the impact of the small sample size for some tissues, in particular the duodenum (one tissue sample). Secondly, for a conservative gene network cluster analysis, the pair-wise gene correlation coefficient cut-off was set to 0.9 and we further filtered transcripts based on relative (fold change) and absolute (counts per million) expression levels. We identified 16 major gene expression patterns, representing common and specific transcriptomic signatures of the epithelial and GI tissues, accounting for 639 different transcripts (Fig. 2A). A full list of the expression of the genes across the tissues with assignment to clusters is available (Table S2, S3). Gene Ontology enrichment analysis of the clusters identified a number of significantly enriched terms (Table 1). A full list of the genes contributing to the enrichments is available (Table S4). Most notable was the highly significant enrichment of the genes in the epithelia-intestine cluster for the GO-term, "cell cycle process". The higher expression of the majority of these genes in the epithelial and GIT tissues (Fig. 2, Table S2) is consistent with the much higher turnover rate of these tissues compared to liver and muscle (Milo et al., 2010) and may contribute to the structural adaptability of the rumen epithelia to different diets and health conditions (Dionissopoulos et al., 2012; Penner et al., 2011). Epithelia structure related pathways including 'cell junctions' showed significant

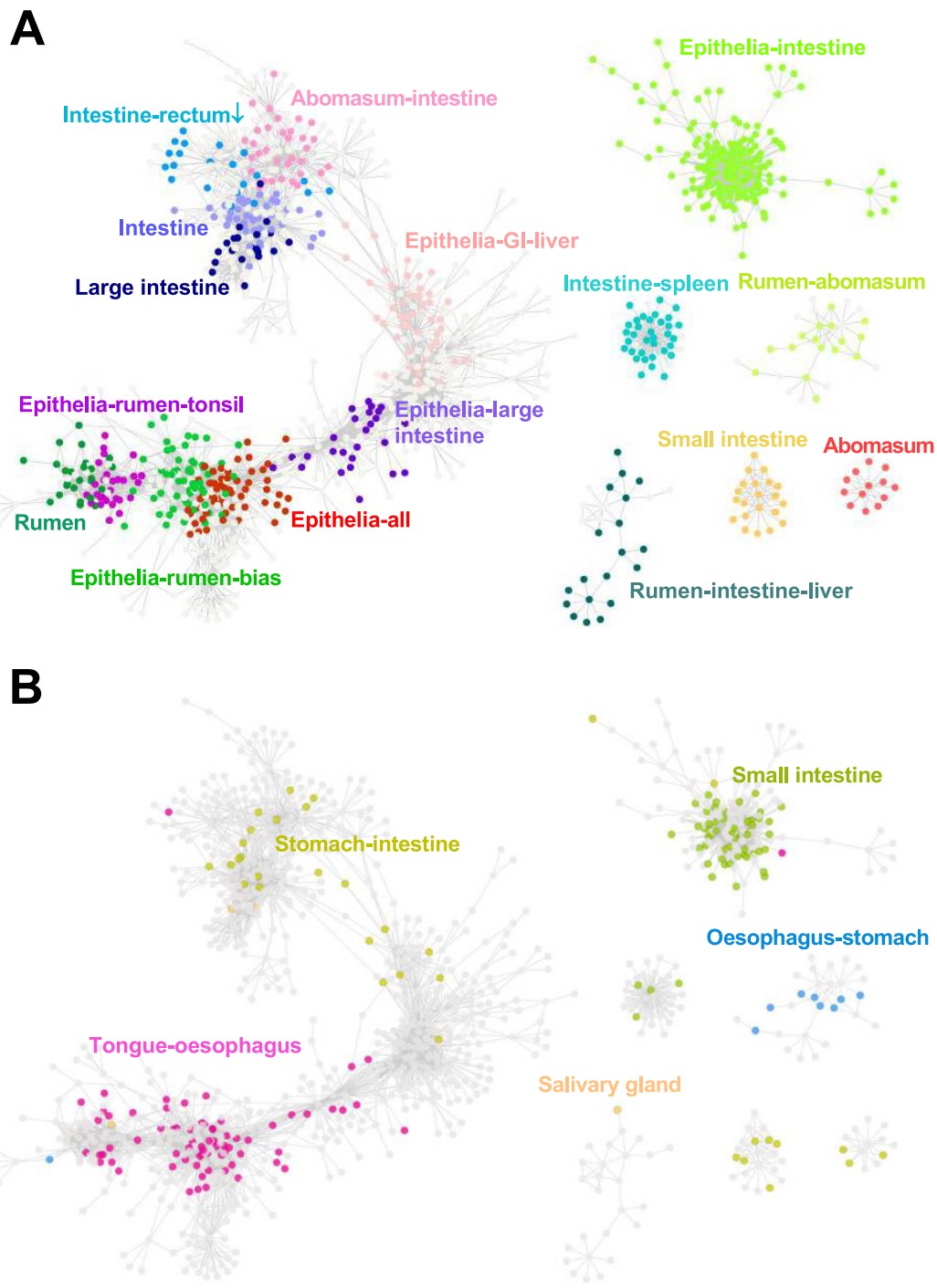

**Figure 2 Gene co-expression network.** (A) Each dot represents a sheep transcript and different colors represent the tissue(s) where the transcript showed high expression, compared to the other tissues. Rectum ↓: low in rectum. (B) The same gene co-expression network with only the orthologous genes present in specific pig GIT clusters (*Freeman et al., 2012*) highlighted (Additional file 1). The names and colors of pig cluster were determined according to the tissues where genes showed the highest and the second highest expression level in the pig GI gene network (*Freeman et al., 2012*).

enrichment in genes highly expressed in the rumen and the large intestine (Table 1). Gene members involved in cell junction functions have been reported to be important for the rumen epithelia to maintain pH homeostasis (*Dionissopoulos et al., 2012*; *Steele et al., 2011a*). Two other very significant enrichments were observed, "flavonoid biosynthetic process" in the rumen-intestine-liver cluster and "regulation of chloride transport" in the large intestine cluster (Table 1). The mammalian Epidermal Development Complex (EDC) locus is a cluster of up to 70 adjacent genes encoding proteins with roles in the development and the structure of stratified epithelia (*Kypriotou, Huber & Hohl, 2012*). Although no significant enrichment of genes in the rumen cluster was identified by GO analysis genes in the EDC region were very significantly overrepresented in the cluster (Table 1). This is consistent with our previous identification of several ruminant specific genes at the EDC locus highly preferentially expressed in the rumen (*Jiang et al., 2014*). The genes in the epithelia-rumen-tonsil cluster were also very significantly enriched for EDC genes (Table 1). Thus the clustering of the rumen with the skin and tonsil appears to have been driven by genes involved in the development and structure of the stratified epithelium.

## The stratified squamous rumen epithelium expression signature

The EDC locus genes are not the only genes encoding proteins involved in the synthesis of the cornified surface of the rumen and we looked for additional genes involved in cornification preferentially expressed in the rumen compared to skin and tonsil. The cross linking of the proteins of the cornified surface is mediated by transglutaminases (TGMs) (*Eckert et al., 2005*). Multiple TGMs are expressed in the rumen in this study, TGM1 and TGM3 appear to be the major rumen transglutaminases, but are also highly expressed in the skin  (Fig. 3). Keratins are major components of the cornified layers so we asked the question, are there keratin genes highly preferentially expressed in the rumen? Although no *KRT* genes showed expression as exclusive to the rumen as some of the EDC locus genes in our data, *KRT36* was grouped in the rumen expression cluster (Fig. 3, Table S3, Fig. S2), with significantly elevated expression in rumen, compared to the other studied tissues, and limited expression in skin. *KRT36* was previously identified as a novel keratin gene only expressed in sheep hair cortex (*Yu et al., 2011*) and its rumen expression showed significant responses to dietary changes in cattle (*Li et al., 2015*). However, in humans the highest expression of *KRT36* was in the tongue (Genevestigator (*Hruz et al., 2008*) analysis). Overall the transglutaminases and keratins do not appear to be as preferentially expressed in the rumen as some of the EDC locus genes.

Kallikrein-related peptidases are involved in the turnover of the cornified layers of the stratified epithelia, and deficiencies can lead to altered turnover of the surface layers of the epithelia (*Hovnanian, 2013*). In our study, *KLK12* is the only *KLK* family member preferentially expressed in the rumen (Fig. 3, Table S2). Members of the SPINK (serine peptidase inhibitor, Kazal type) family are inhibitors of the KLK family peptidases (*Hovnanian, 2013*), *SPINK5* is the only member of the family that is highly expressed in the rumen (Fig. 3, Table S2) in our data, but is also highly expressed in the tonsil and skin.
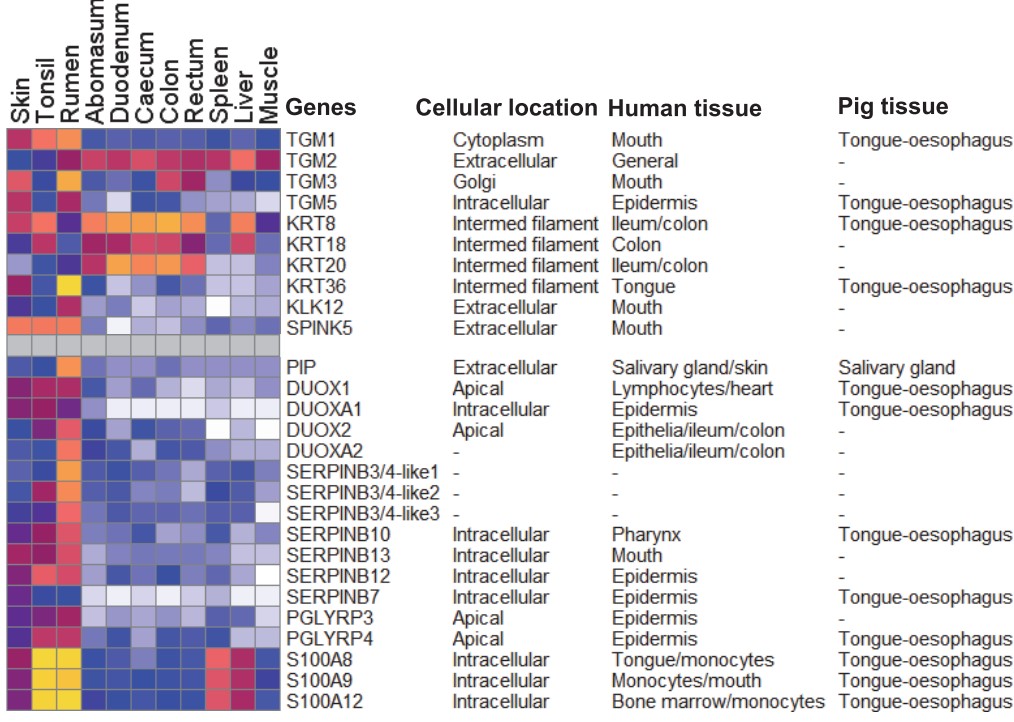

**Figure 3  Expression profiles of innate immunity and epithelial development genes in sheep.** Data are presented with log₂ Fragments Per Kilobase of exon per Million fragments mapped (FPKM) values along with the subcellular locations and/or tissues of pig (*Freeman et al., 2012*) and human (Genevestigator (*Hruz et al., 2008*) analysis) where these genes showed high expression. Cellular location information were derived from GENATLAS database (*Frezal, 1998*).

KLK12 and SPINK5 may be involved in the regulation of the turnover and thickness of the cornified surface of the rumen epithelium, but may not form a rumen specific system.

## Rumen micro-organism interactions

The rumen is the site of frequent interaction between the host and very dense populations of micro-organisms. In our study, *DUOX2* and *DUOXA2* encoding subunits of dual oxidase were preferentially expressed in the rumen (Fig. 3), while *DUOX1* showed rumen-biased expression (Fig. 3) and *DUOXA1* was highly expressed in all epithelia tissues (Fig. 3). This observation is in line with the findings in the pig where the highest expression of *DUOXA1* and *DUOX1* was in the epithelial tissues, e.g., tongue and lower oesophagus (*Freeman et al., 2012*). In humans, the *DUOXA1* and *DUOX1* genes are also most highly expressed in epithelia tissues exposed to air, whilst *DUOX2* and *DUOXA2* are most highly expressed in a different set of tissues including the GIT (Genevestigator (*Hruz et al., 2008*) analysis). Thus, our findings suggest that the DUOX1s are active in general epithelial tissues, while DUOX2s are probably active specifically in rumen to play a major role in controlling microbial colonization. Previously in sheep, the highest expression levels of *DUOX1* and

*DUOX2* were reported in the bladder and abomasum, respectively, but the rumen and epithelial tissues were not included in the tissues surveyed (*Lees et al., 2012*).

*PIP*, encoding prolactin-induced protein (an aspartyl protease), was preferentially expressed in the rumen (Fig. 3). In humans, *PIP* is also highly expressed in epidermal (Genevestigator (*Hruz et al., 2008*) analysis) and exocrine tissues, and in pigs in the salivary gland. Although PIP has been reported to be involved in regulation of the cell cycle in human breast epithelial cells (*Cassoni et al., 1995*; *Naderi & Vanneste, 2014*), its expression pattern in sheep (not part of the cell cycle cluster) is more consistent with a role in mucosal immunity (*Hassan et al., 2009*). Also highly expressed in the rumen were members of the *SERPINB* family of peptidase inhibitors (Fig. 3), which are involved in the protection of epithelial surfaces in humans (*Wang et al., 2012*) and mice (*Sivaprasad et al., 2011*). EDC locus genes *PGLYRP3* and *PGLYRP4* encode peptidoglycan recognition proteins in the N-acetylmuramoyl-L-alanine amidase 2 family, which bind to the murein peptidoglycans of Gram-positive bacteria as part of the innate immune system. Additional EDC locus genes, *S100A8*, *S100A9* and *S100A12* (calgranulins A, B and C), encode key players in the innate immune function (*Funk et al., 2015*; *Tong et al., 2014*).

## Rumen steroid metabolism

Amongst the genes preferentially expressed in the rumen (and often the liver) we identified a number of aldo/keto-reductases (Fig. 4). *AKR1C1* can catalyze the conversion of progesterone to 20-alpha-hydroxy-progesterone (PGF2$\alpha$) (*Penning, 1997*), retinals to retinols and bioactivates and detoxifies a range of molecules (*El-Kabbani, Dhagat & Hara, 2011*). Intravenous injection of PGF2$\alpha$ in goats has been shown to increase contraction strength of rumen smooth muscle, which leads to a reduction in the contraction rate of the rumen (*Van Miert & Van Duin, 1991*; *Veenendaal et al., 1980*). *AKR1C1* has also been reported to be preferentially expressed in the rumen of cattle (*Kato et al., 2015*). The exact role of AKR1C1 in the rumen is unknown. In addition, the gene encoding the related enzyme AKR1D1 (catalyzes the reduction of progesterone, androstenedione, 17-alpha-hydroxyprogesterone and testosterone to 5-beta-reduced metabolites) is highly expressed in the rumen and the liver and the gene encoding ARK1C4 in the rumen, liver and duodenum (Fig. 4). The products of these genes are also likely to be involved in the metabolism of steroids in the rumen epithelium. In addition, we observed marked pathway enrichment of flavonoid biosynthetic process due to the identification of five members of the UDP-glucuronosyltransferase (UGT) gene family (*Well et al., 2004*), with the highest expression levels in the rumen and liver (Table S2). Flavonoids are only produced by plants, but UGT enzymes are highly active in mammals and catalyze the glucuronidation of a diverse chemical base including steroids, bile acids and opioids (*Well et al., 2004*). The functions of the products of these genes in the rumen require further investigation. However, results discussed here suggest important interactions between the rumen wall and activity of steroids.

## Comparison of the sheep and pig GIT transcriptomes

To compare the ruminant and a closely related non-ruminant mammal GIT transcriptomes (*Jiang et al., 2014*), we mapped those transcripts previously reported to show specific
**Table 2  Representation of the pig GIT gene clusters in the sheep GIT network.**

| Pig cluster[a] | Pig tissues[a] | Pig cell type of origin[b] | Overlap | P-value[b] | Representation | Sheep tissues | Go term enrichment | P-value[c] |
|---|---|---|---|---|---|---|---|---|
| Overall | | | 179 | 8.1E–31 | Over | | Cell cycle process | 2.0E–13 |
| 1, 7 | Intestine | Immune cells/cell cycle | 58 | 2.4E–11 | Over | Epithelia, intestine | Cell cycle process | 1.5E–33 |
| 3, 8 | Tongue-oesophagus | Stratified squamous epithelia | 73 | 1.3E–34 | Over | Rumen, epithelia, abomasum, large intestine | Epidermis development | 2.9E–05 |
| 2, 4, 9 | Oesophagus-stomach | Muscle | 9 | 0.0002[d] | Under[d] | Rumen, abomasum | na | |
| 6, 13, 15 | Salivary gland | Stratified columnar epithelia | 4 | 0.1777 | None | | na | |
| 5, 12, 14, 16 | Stomach-intestine | Ciliate/glandular epithelia | 35 | 5.4E–09 | Over | Stomach intestine | na | |
| 10 | Stomach | Neuronal | 0 | na | na | | na | |

**Notes.**

[a]Numbers, names and grouping of pig gene clusters by cell type of origin are according to *Freeman et al. (2012)*.

[b]Calculated hypogeometric *P* values, representing the significance of representation of pig genes in sheep gene network.

[c]FDR corrected GO term enrichment *P* values.

[d]If overlap with just the rumen and rumen-abomasum clusters, significant ($P = 8E–05$) over representation.

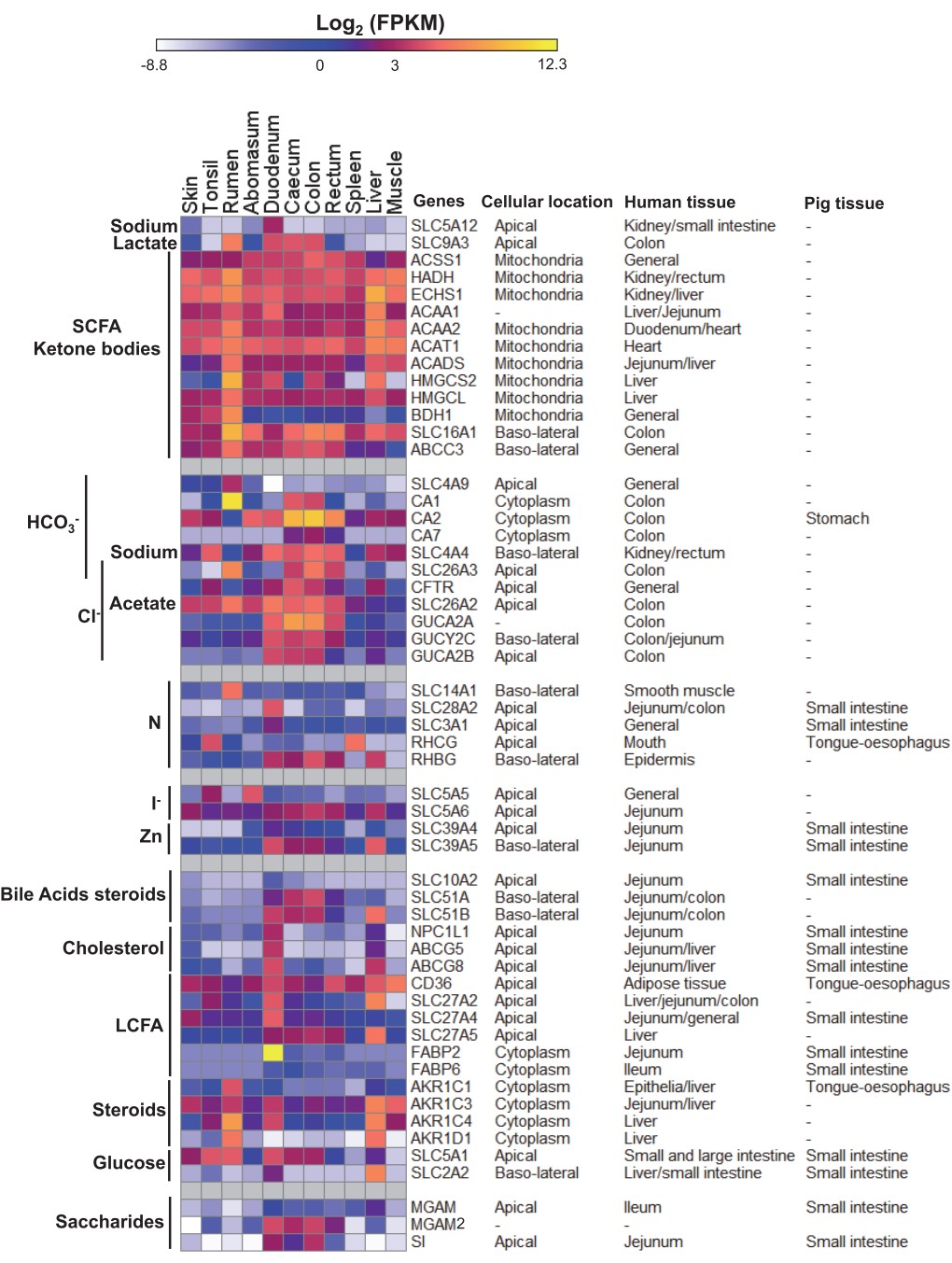

**Figure 4  Gene expression profiles of metabolic processes discussed in the text.** Data are presented with log$_2$ Fragments Per Kilobase of exon per Million fragments mapped (FPKM) values along with the subcellular locations and/or tissues of pig (*Freeman et al., 2012*) and human (Genevestigator (*Hruz et al., 2008*) analysis) where these genes showed high expression. Texts and bars on the left side of the heatmap indicate involved pathways for covered genes described in the article. Cellular location information were derived from GENATLAS database (*Frezal, 1998*).

expression patterns in the pig GIT (*Freeman et al., 2012*) to the sheep gene network (Fig. 2B). Pig is the genomically closest non-ruminant to the ruminants (*Groenen et al., 2012*; *Jiang et al., 2014*) for which sufficient GIT transcriptome data is available. The overall overlap of the 639 genes in the sheep GIT network and the 2,634 mappable pig GIT genes is 179, which is highly significant (Table 2). The smaller number of genes showing differential expression in our study versus the pig study is due to the application of stringent statistical filtering thresholds to minimize the impact of the small number of samples per tissue. However, the overlap of 627 genes between the set of 2,475 sheep genes identified using relaxed filtering criteria and the 2,634 pig genes was also highly significant ($P < 10E-20$), supporting the robustness of the approach. The set of 179 overlapping genes was highly significantly enriched for the GO-term "cell cycle process" (Table 2). The overlap of genes between the pig intestine clusters and the sheep epithelia-intestine cluster was highly significant and the overlap genes were again very highly significantly enriched for the GO-term "cell cycle process" (Table 2). A full list of the genes in the overlap and assignment to the pig and sheep gene clusters is available (Table S5). Furthermore, pig genes preferentially expressed in the tongue and oesophagus have a highly significant overlap with sheep genes with high expression in the rumen and epithelial tissues (Fig. 2B), enriched for the GO-term "epidermis development" (Table 2). Our results emphasises the contribution of cell cycle to the renewal of mammalian GIT epithelial surfaces (*Crosnier, Stamataki & Lewis, 2006*).

## Ruminant specific pathways for SCFA uptake and GIT metabolism?

SCFAs are the major source of energy in ruminants, with the primary sources of SCFAs being the rumen, and to a much lesser extent the large intestine. Carbonic anhydrases, which hydrate $CO_2$ to bicarbonate, are thought to play a significant role in the uptake of SCFAs by an SCFA/bicarbonate antiporter, and by providing protons at the rumen epithelium to neutralize the SCFAs and promote their diffusion into the ruminal epithelium (*Bergman, 1990*; *Wang, Baldwin & Jesse, 1996*). There are many members of the carbonic anhydrase gene family (*Tashian, 1989*), several of which are expressed in mammalian gastrointestinal tissues (*Freeman et al., 2012*; *Kivel et al., 2005*; *Parkkila et al., 1994*; *Tashian, 1989*). In ruminants, *CA1* has previously been reported to encode a rumen specific carbonic anhydrase with low activities in the blood (unlike in other mammals) and in the large intestines (*Carter, 1971*). Consistent with this, compared to all of the other tissues in our dataset, *CA1* is highly expressed in the rumen and, albeit with lower but significant expression, in the large intestine (Fig. 4). *CA2* and *CA7* appear to encode the major carbonic anhydrases in the large intestines (Fig. 4). In humans *CA1*, *CA2* and *CA7* are highly expressed in the colon (Genevestigator (*Hruz et al., 2008*) analysis). In contrast in pigs, whilst *CA2* is highly expressed in the stomach, it is not highly expressed in the large intestine and *CA1* and *CA7* were not reported to be differentially expressed across the GIT (*Freeman et al., 2012*).

The apical membrane SCFA/bicarbonate antiporter exchanges intracellular bicarbonate with intra-ruminal SCFA and consistent with previous publications, *SLC4A9*, preferentially expressed in the rumen in our dataset (Fig. 4), encodes the most likely antiporter. The proposed basolateral membrane SCFA/bicarbonate antiporter gene *SLC16A1* (exchanges

intracellular SCFA with blood bicarbonate), which has highest expression in the rumen in our dataset, followed by the colon and rectum, has a much more general expression across the tissues than *SLC4A9* (Fig. 4). These expression patterns are consistent with previous findings in cattle (*Connor et al., 2010*). SLC16A1 is also likely to be involved in the transport of ketone bodies into the blood supply to the basolateral surface of the rumen epithelium (*Van Hasselt et al., 2014*).

$HCO_3^-$-independent apical uptake of acetate from the rumen has also been observed (*Aschenbach et al., 2009*). However, the transporter has not been identified, with candidates proposed in the SLC4A, SLC16A, SLC21A, SLC22A and SLC26A families (*Aschenbach et al., 2009*). Members of the SLC21A and SLC22A families showed generally low expression in the rumen in our study (Table S2). In addition to *SLC16A1* and *SLC4A9* discussed above, *SLC26A2* and *SLC26A3* are highly expressed in the rumen in our dataset (Fig. 4). Both genes encode apical anion exchangers confirming them as candidates for encoding the apical $HCO_3^-$-independent acetate uptake transporter. SLC26A3 is a $Cl^-/HCO_3^-$ exchanger (see fluid and electrolyte balance section below) and therefore is unlikely to be an $HCO_3^-$-independent acetate transporter. However, SLC26A2 is a $SO_4^{2-}/OH^-/Cl^-$ exchanger (*Ohana et al., 2012*) and remains a candidate for the proposed apical $HCO_3^-$-independent acetate transporter. An $HCO_3^-$-independent basolateral maxi-anion channel for $SCFA^-$ efflux to blood has also been proposed without an assigned transporter (*Georgi et al., 2014*). A survey of ABC (ATP-binding cassette) family transporters identified *ABCC3* as the most preferentially expressed in the rumen in our dataset and with the second highest expression in the large intestine (Fig. 4). ABBC3 is an organic anion transporter with a possible role in biliary transport and intestinal excretion (*Rost et al., 2002*). Therefore, *ABCC3* may be involved in the efflux transport of $SCFA^-$ from the rumen epithelium to blood.

In most mammals, including humans, the liver is the major site of the synthesis of ketone bodies (acetoacetate and beta-hydroxybutyrate), but in ruminants the epithelium of the rumen is a major site of *de novo* ketogenesis (*Lane, Baldwin & Jesse, 2002*). *HMGCS2* encodes an HMG-CoA synthase (3-hydroxy-3-methylglutaryl-CoA Synthase 2) in the ketogenesis pathway (Fig. 5). This gene is significantly associated with cattle butyrate metabolism (*Baldwin et al., 2012*) and the encoded enzyme was predicted to be the rate limiting enzyme in sheep ruminal ketone body synthesis (*Lane, Baldwin & Jesse, 2002*). As expected, in our data *HMGCS2* is highly expressed in the rumen compared to the other GIT tissues and the liver (Fig. 4). *ACADS*, *HMGCL* and *BHD1*, which encode other enzymes involved in the ketone body pathway (Fig. 5), are also highly expressed in the rumen relative to most of the other tissues studied (Fig. 4). HMGCS1 and ACAT2 may also contribute to the ketone body pathway in the rumen, but their highest expression levels are in the liver (Table S1). However, their expression in the rumen has been reported to actively respond to different diets (*Steele et al., 2011b*) and acidosis conditions (*Steele et al., 2012*) in cattle. Whilst *HMGCS2* is quite highly expressed in the colon, in contrast *ACADS*, *HMGCL* and *BHD1* are not highly expressed (Fig. 4), consistent with the colon not being a major contributor to ketone body synthesis. Genes encoding enzymes for other steps in the pathways from acetate and butyrate to ketone bodies are much more generally expressed across the tissues, although expression of *ECHS1* and *ACAT1* are significantly higher in

the rumen than in other GIT tissues (Fig. 4). In humans, in addition to the liver, *HMGCS2* also has high expression in the intestine, including the jejunum and colon (Genevestigator (*Hruz et al., 2008*) analysis). In contrast, the only enzyme in the pathway (Figs. 4 and 5) reported to be preferentially expressed in the pig GIT was *BDH1*, in the fundus of the stomach (Table S5). Thus the rumen, abomasum, duodenum, caecum, colon and rectum in sheep all appear to have subtly different SCFA transport and metabolism systems, and in the equivalent compartments of the GIT appear to be different between sheep, humans and pigs.

## Long chain fatty acids (LCFAs) uptake, cholesterol homeostasis and bile acid recycling

Due to the activity of the microbial populations of the rumen and the production of SCFAs ruminants have less reliance on dietary LCFAs than non-ruminants. Does this reduced importance lead to detectable differences in the transcriptome? The small intestine is the principal site of uptake of LCFA and cholesterol homeostasis, and consistent with this the genes encoding the well characterized components of the intestinal fatty acid uptake (*CD36*, *SLC27A2/4/5* and *FABP2* (*Wang et al., 2013*)) and cholesterol homeostasis (*NPC1L1* and *ABCG5/8* (*Wang et al., 2013*)) systems are expressed in the sheep small intestine (Fig. 4), as they are in humans and most are in the pig (*Freeman et al., 2012*). *FABP2* and *ABCG5* are particularly preferentially expressed in the sheep small intestine relative to other GIT tissues (Fig. 4). However, it is thought that the major route of LCFA uptake at the apical membrane of the GIT epithelium is by passive diffusion (*Abumrad & Davidson, 2012*).

Bile acids secreted by the liver and stored in the gall bladder before being released into the small intestine play a major role in the uptake of LCFAs. Bile acids are recycled in the intestine. SLC10A2 in the apical membrane and SLC51A and SLC51B in the basolateral membrane are proposed to constitute the uptake systems in the human small intestine (*Ballatori et al., 2013*). *SLC10A2* is also preferentially expressed in the small intestines of the pig, but preferential expression of *SLC51A/B* has not been reported (*Freeman et al., 2012*). In sheep *SLC10A2* is preferentially expressed in the small intestine, albeit it a low level (Fig. 4). Whilst *SLC51B* is highly expressed in the duodenum in sheep, the highest expression of the two subunits together in sheep (*SLC51A/B*) is in the caecum and the colon (Fig. 4), where they are also expressed in humans and mice (Genevestigator (*Hruz et al., 2008*) analysis). Although described as subunits of a complex, *SLC51A* and *SLC51B* have also been reported to be regulated differently (*Ballatori et al., 2013*), thus the balance between expression of *SLC10A2* and *SLC51A* and *SLC51B* may indicate differences in the bile acid uptake pathways in the duodenum, large intestines and liver of sheep.

Overall despite the reduced importance of LCFAs sheep appear to have a very similar systems to human and pigs for LCFA uptake and bile acid recycling.

## Saccharide metabolism

Again as a consequence of the activity of the rumen microbes in mature ruminants the uptake of dietary glucose may be less than 10% of glucose requirements (*Young, 1977*). The dietary glucose comes primarily from the degradation of polysaccharides, in particular

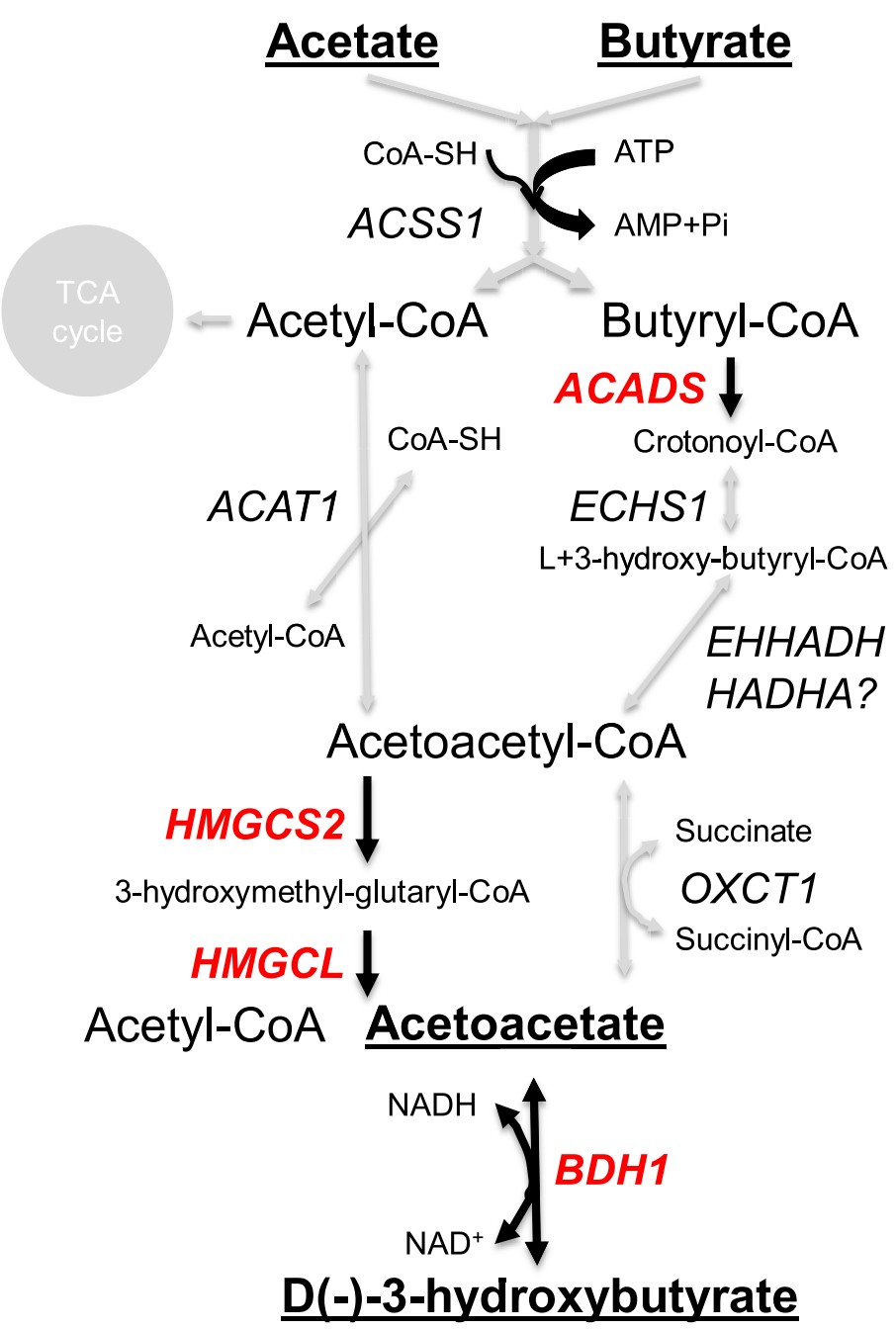

**Figure 5** **Ruminant ketone body metabolism pathways.** Key enzyme encoding genes (red text) and pathways (black arrow) are highlighted.

in the small intestine of starch that has escaped degradation by the rumen microbial population. The primary source of alpha-amylase required to digest the long polymers is the pancreas, which was not investigated in this study. Genes encoding three enzymes likely to contribute to the digestion of starch and other alpha-glycans, *MGAM* (maltase-glucoamylase), *MGAM2* (maltase-glucoamylase 2) and *SI* (sucrase-isomaltase) (*Nichols et al., 2003*), were preferentially expressed in the tissues studied here. *SI* was preferentially

expressed in the intestine-low in rectum gene cluster, *MGAM2* was highly expressed in all intestinal tissues, while *MGAM* was also preferentially expressed in the intestine (primarily the duodenum), but at a much lower level (Fig. 4). In humans (Genevestigator (*Hruz et al., 2008*) analysis) and pigs (*Freeman et al., 2012*), both *MGAM* and *SI* are preferentially expressed in the small intestine. Expression of the orthologues of *MGAM2* has not been reported in the GIT of humans (Genevestigator (*Hruz et al., 2008*) analysis) or pigs (*Freeman et al., 2012*).

The mammalian *MGAM* and *MGAM2* genes appear to have arisen by tandem duplication of a single ancestral gene at the base of the mammals (*Nichols et al., 2003*; *Nichols et al., 1998*). *MGAM2* genes are present in most mammals, but are not well characterized in any species. Comparative analysis of the protein sequences of MGAM and MGAM2 showed that MGAM2 has additional sequence at the carboxy-terminus comprised of multiple copies of a 40 amino acid repeat not present in MGAM (Fig. 6). The repeat unit is enriched in serine and threonine, with similar sequences in the predicted sheep, cattle, pig and to a much lesser extent human proteins (Fig. 6). The repeat unit of MGAM2 is predicted to be heavily glycosylated (analysis of *Steentoft et al. (2013)*) to form a mucin-like domain. As in the rumen, the microbial population in the large intestine ferments plant material, contributing up to 10% of the total carbohydrate fermentation and conversion to SCFAs in the ruminant GIT (*Gressley, Hall & Armentano, 2011*). Whilst the role of MGAM2 is unclear it appears to represent a contribution from the host to the breakdown of plant polysaccharides by the bacterial population in the large intestine. MGAM produces glucose from maltose and MGAM2 may have a similar functionality, and therefore contribute to the uptake of the scarce supply of glucose in ruminants. Alternatively the high expression of MGAM2 and low expression of MGAM may reflect the reduced availability of glucose in the rumen GIT. Further investigation of this gene and the activity and function of its encoded protein will improve our understanding of carbohydrate metabolism in the large intestine of ruminants.

In humans the major uptake of glucose in the GIT occurs in the small intestine via SLC5A1 (aka SGLT1) in the apical membrane, and SLC2A2 (aka GLUT2) in the basolateral membrane (*Roder et al., 2014*). The expression pattern of these two genes in sheep (Fig. 4) and pigs (*Freeman et al., 2012*) is consistent with a similar process in all three species.

## Nitrogen acquisition and recycling

A high level of nitrogen recycling in the GIT is a characteristic of ruminants. Urea is the major input from the animal (primarily via the saliva and the rumen epithelium) and anabolic-N sources (in the small intestine) and ammonia (in the rumen, small and large intestines) are the major uptake molecules from the GIT (*Lapierre & Lobley, 2001*). *SLC14A1* (Fig. 4), encoding SLC14A1 which mediates the basolateral cell membrane transport of urea, a key process in nitrogen secretion into the GIT (*Abdoun et al., 2010*), is highly preferentially expressed in the rumen in our dataset (Fig. 4). However, in cattle expression of *SLC14A1* was not affected by differences in dietary N (*Rojen et al., 2011*) and doubts remain about the role of SLC14A1 in increasing rumen epithelial urea permeability at low dietary N. Urea is also thought to be released by the epithelium of the small and

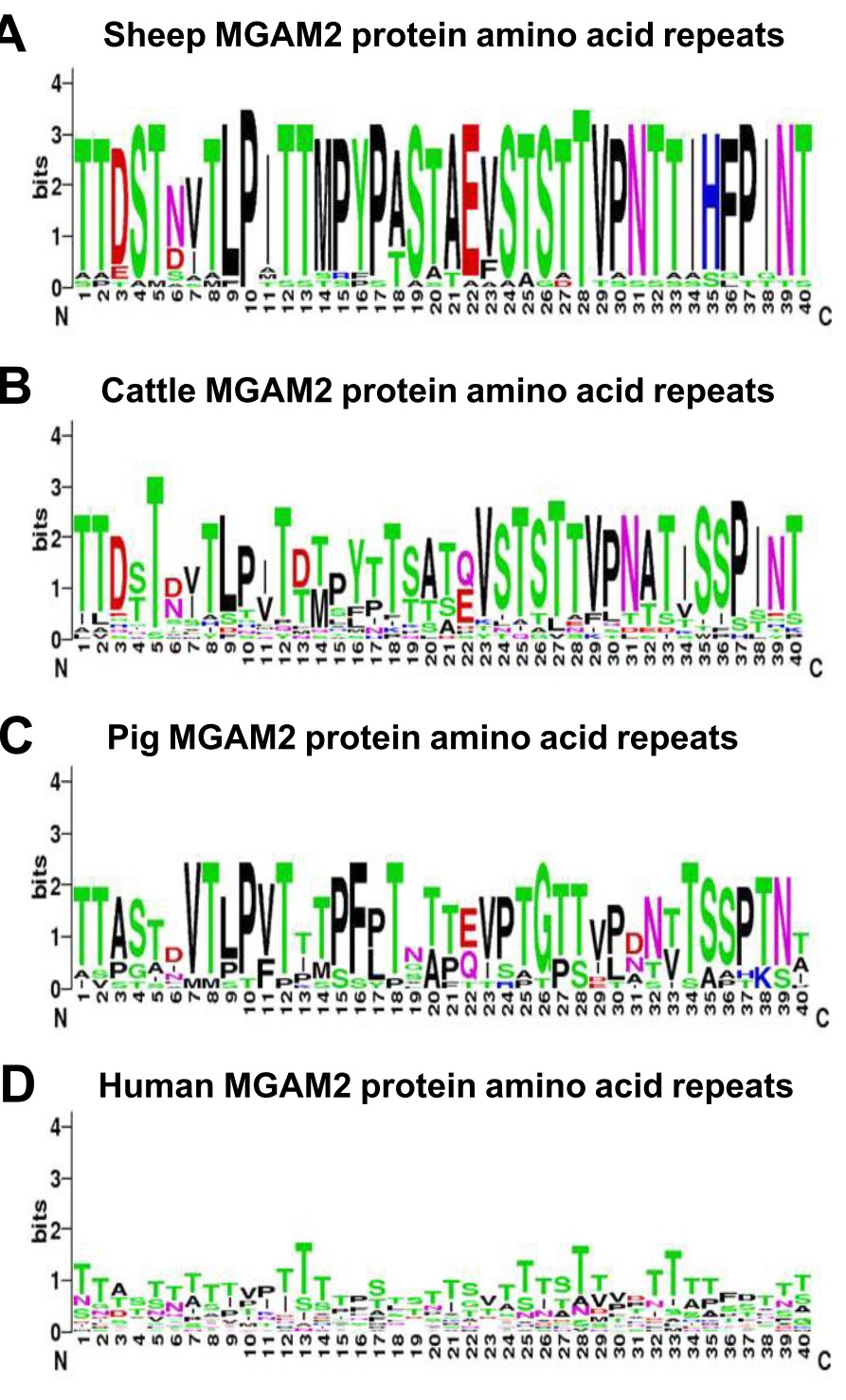

**Figure 6 Organization of the MGAM2 carboxy-teminus.** Consensus motifs of the serine/threonine rich 40 amino acid repeats at the carboxy-terminus of predicted MGAM2 proteins. (A) sheep. (B) cattle. (C) Pig. (D) human.

large intestines (*Lapierre & Lobley, 2001*), but our analysis did not identify a potential transporter.

Urea is converted to ammonia by microbial ureases and is used by rumen microorganisms to synthesize microbial proteins (75–85% of microbial N) and nucleic acids (15–25% of microbial N) (*Fujihara & Shem, 2011*) which are subsequently digested by the host in the intestines, thus recovering the majority of the secreted nitrogen (*Abdoun, Stumpff & Martens, 2006*). Consistent with this, *SLC3A1* (neutral and basic amino acid transporter) in our study is preferentially expressed in the duodenum (Fig. 4), as is *SLC28A2* (concentrative nucleoside transporter) the product of which plays an important role in intestinal nucleoside salvage and energy metabolism (*Huber-Ruano et al., 2010*). Both genes were also highly expressed in the small intestine of pigs (*Freeman et al., 2012*) and humans (Genevestigator (*Hruz et al., 2008*) analysis). *RHBG* (*SLC42A2*), an ammonia transporter, is preferentially expressed in the sheep small and large intestines and the liver (Fig. 4) and is a candidate for an intestinal ammonia transporter. However, *RHBG* is not expressed at particularly high levels in the human GIT (Genevestigator (*Hruz et al., 2008*) analysis) relative to many other tissues, and was not reported to be preferentially expressed in the pig GIT (*Freeman et al., 2012*). In humans uptake of ammonia in the large intestine is thought to most likely occur (mainly) by passive non-ionic diffusion (*Wrong & Vince, 1984*). However, *RHCG* (apical membrane) and *RHBG* (basolateral membrane) have also been proposed to constitute an ammonium uptake pathway in the human GIT (*Handlogten et al., 2005*). The expression profile of *RHCG* in sheep (Fig. 4) is not consistent with such a pathway in sheep.

In addition to the secretion of urea into the rumen (a ruminant specific process) the increased importance of nitrogen recycling in ruminants may have led to the apparent increased expression of *RHBG* in the GIT of sheep.

## Iodine recycling

*SLC5A5*, member 5 of solute carrier family 5, encoding a sodium iodide symporter is highly preferentially expressed in the abomasum in our study (Fig. 4). *SLC5A5* also has higher expression in human (Genevestigator (*Hruz et al., 2008*) analysis) and rat stomach (*Kotani et al., 1998*) than in other digestive tissues. The latter authors reported that the distribution of SLC5A5 transcripts in the stomach epithelium was consistent with a role of SLC5A5 in the import or export of iodine, from or to the stomach contents. In the rat, iodine is actively transported into the gastric lumen and this transport is at least partly mediated by a sodium-iodide symporter (*Josefsson et al., 2006*). In cattle the rate of iodine export by the abomasum epithelium into the abomasum is much greater than the import of iodine from the abomasum (*Miller, Swanson & Spalding, 1975*), suggesting that the role of SLC5A5 in sheep abomasum is to export iodine into the stomach contents. In contrast, *SLC5A5* was not reported to be significantly more expressed in the pig stomach versus other components of the GIT (*Freeman et al., 2012*). The specific physiological role of iodine in the stomach/GIT is unknown, but a number of possibilities have been suggested: iodine-conserving mechanisms to deal with low iodine concentrations in the diet (*Miller, Swanson & Spalding, 1975*), antioxidative activity (*Venturi & Venturi, 1999*) and

antimicrobial activity (*Spitzweg et al., 1999*). The majority of the secreted iodine is thought to be recovered in the lower intestines. Another member from the same transporter family SLC5A6, a sodium/multivitamin and iodide co-transporter (*De Carvalho & Quick, 2011*), encoded by a gene showing expression in all studied tissues, with the highest expression sheep large intestine (Fig. 4) is a likely candidate for the iodine importer. In humans, *SLC5A6* is also expressed in a wide range of tissues with intestinal tissues being close to the top of the list (*De Carvalho & Quick, 2011*). In pigs, *SLC5A6* is preferentially expressed in the small intestine (*Freeman et al., 2012*). The high expression of *SLC5A5* in the abomasum suggests that ruminants may have retained a higher dependence on iodine in the GIT than other mammals.

### Zinc homeostasis

*SLC39A4* encodes a transporter protein essential for zinc uptake in the mouse intestine (*Dufner-Beattie et al., 2003*) and stomach (*Martin et al., 2013*). *SLC39A4* is highly expressed in stomach and intestines in sheep (Fig. 4) and humans (Genevestigator (*Hruz et al., 2008*) analysis), and showed the highest expression in pig small intestine (*Freeman et al., 2012*). Another zinc transporter encoding gene, *SLC39A5*, has a similar expression profile to *SLC39A4* in sheep (Fig. 4), humans and pigs. However, SLC39A5 is located in the basolateral membrane and is involved in the secretion of zinc. In mouse gastrointestinal tract cells the two zinc transporters are reciprocally regulated (*Weaver et al., 2007*), together controlling the influx and efflux of zinc at the intestinal epithelium. It appears likely that sheep have a similar mechanism for zinc homeostasis to other mammals.

### Fluid and electrolyte balance

Maintaining salt and water balance is an important function of the mammalian GIT. In the large intestine significant GO term enrichment was identified for regulation of chloride transport, due to the inclusion of *CA2*, *7* and *CFTR* (Table 1, Fig. 4). This is in agreement with the reported critical chloride secretory mechanism in intestinal epithelial cells, associated with mucosal hydration (*Barrett & Keely, 2000*). SLC26A3, which is a $Cl^-/HCO_3^-$ antiporter, imports $Cl^-$ ions driven by bicarbonate, thus linking the activity of carbonic anhydrases and the leakage of $Cl^-$ out of the cells by CFTR. *SLC26A3* is preferentially expressed in the large intestine of sheep (Fig. 4) and the colon of pigs. Thus the expression of genes involved in fluid and electrolyte balance is similar between all three species.

## CONCLUSIONS

As a significant event in the evolution of the true ruminants, the evolutionary origin of the rumen is the subject of debate, with out-pouching of the oesophagus, or of the stomach, as the two most likely origins (*Beck, Jiang & Zhang, 2009*; *Langer, 1988*). The cornification of the epithelia surface, tissue clustering analysis based on gene expression (driven by the epidermal structural proteins and innate immunity genes) and the relative lack of metabolic overlap with the abomasum strongly favours an oesophageal origin. Metabolically the rumen has many similarities with the liver, especially for SCFA metabolism and even

though there are functional similarities with the large intestine, the complements of genes involved are not highly similar.

We have identified a small number of highly rumen specific metabolic processes, in particular the roles of SLC14A1 (urea secretion), SLC4A9 (SCFA uptake) and AKR1C1 (uncertain function). Overall our analysis has enabled gene expression data to be married up with decades of physiological and other research to link transport and enzymatic activities and the most likely genes encoding products with the activities. Nitrogen and iodine recycling have been identified as processes with a much greater importance in the sheep than in humans or pigs. These metabolic functions are protected by strong immune functions and stratified epidermis-like epithelium. The major rumen immune players are DUOX and SERPINB gene families and *DUOXA2*, *DUOX2s* and *SERPINB3/4-like 1* appear to be preferentially expressed in the rumen. These findings will bring novel insights into biomedical research on mammalian digestive and gastrointestinal systems.

## ACKNOWLEDGEMENTS

We would like to thank Richard Talbot for supervision of the generation of the RNA-Seq data.

### Funding

This work was supported by the Department of Agriculture, Filling the Research Gap "International Coordination of the Rumen Pangenome project" FTRG-1194147-75. ALA and RT acknowledge support from BBSRC Institute Strategic Programme Grants. The Ensembl annotation was funded by BBSRC BB/I025328/1. Sequencing was carried out by Edinburgh Genomics, The University of Edinburgh. Edinburgh Genomics is partly supported through core grants from NERC (R8/H10/56), MRC (MR/K001744/1) and BBSRC (BB/J004243/1). The funders had no role in study design, data collection and analysis, decision to publish, or preparation of the manuscript.

### Grant Disclosures

The following grant information was disclosed by the authors:
Department of Agriculture, Filling the Research Gap "International Coordination of the Rumen Pangenome project": FTRG-1194147-75.
BBSRC Institute Strategic Programme Grants.
BBSRC: BB/I025328/1.
NERC: R8/H10/56.
MRC: MR/K001744/1.
BBSRC: BB/J004243/1.

### Competing Interests

The authors declare there are no competing interests.

## Author Contributions

- Ruidong Xiang conceived and designed the experiments, performed the experiments, analyzed the data, wrote the paper, prepared figures and/or tables, reviewed drafts of the paper.
- Victor Hutton Oddy and Phillip E. Vercoe wrote the paper, reviewed drafts of the paper, contribution of interpretation of results.
- Alan L. Archibald analyzed the data, contributed reagents/materials/analysis tools, wrote the paper, prepared figures and/or tables, reviewed drafts of the paper, contribution of interpretation of results.
- Brian P. Dalrymple conceived and designed the experiments, performed the experiments, analyzed the data, contributed reagents/materials/analysis tools, wrote the paper, prepared figures and/or tables, reviewed drafts of the paper.

## Data Availability

The research in this article used published datasets (European Nucleotide Archive study accession PRJEB6169) and did not generate any raw data. The major secondary datasets are included in the Supplemental Information.

## Supplemental Information

Supplemental information for this article can be found online at http://dx.doi.org/10.7717/peerj.1762#supplemental-information.

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
