# Peer review of "Epithelial, metabolic and innate immunity transcriptomic signatures differentiating the rumen from other sheep and mammalian gastrointestinal tract tissues"

_PeerJ, doi:10.7717/peerj.1762_

## Round 0.1 · original submission · Minor Revisions

· Academic Editor

Minor Revisions

The referees raised several important points that you need to consider and address. In particular, it is important that you explicitly address how the analyses performed in this manuscript are unique from those in the paper "The Sheep Genome Illuminates Biology of the Rumen and Lipid Metabolism" (DOI: 10.1126/science.1252806), as mentioned by one of the reviewers. I am also interested in your response to the question of organism choice - why were pig and human used? Simply because they had the best expression and annotation information available?

In addition to the referees' comments, can you please address the following:

l. 124 - Is Love et al. 2013 the best citation for EdgeR?

l. 171: Please include citations for GLay and Cytoscape.

l. 531 - change involved to involving

Does Table 1 include all of the signficantly enriched terms for each cluster, or only select ones?

Figures 3 and 4: Please explain in the legend where the 'cellular location' information was derived from.

Figure 4: Please explain in the figure legend what the bars and text on the left hand side of the figure are supposed to represent.

I couldn't find legends for the supplementary tables.

Figure S1. In the figure legend, please provide descriptions distinguishing panels A and B. Also, the labels within the panels are hard to read.

·

Basic reporting

Ruminants are very successful herbivorous mammals, in part due to their specialized forestomachs, the rumen complex, which facilitates the conversion of feed to soluble nutrients by micro-organisms. The study systematically analyzed gene expression data through 11 tissues or cell-types and compared them to pig and human, which answered three questions basically the authors raised at the beginning. This manuscript obtained relative good results and can provide valuable information for ruminant studies.

Experimental design

The authors did expression clusters by tissues and by transcripts, respectively, to identify relationships between tissues/cells.

Validity of the findings

No Comments.

Additional comments

1. You used the data downloaded from the paper "The Sheep Genome Illuminates Biology of the Rumen and Lipid Metabolism", that developed and analyzed a high quality reference sheep genome and transcriptomes from 40 different tissues, besides, they also studied the rumen evolution and lipid metabolism by gene expression in ruminants compared to non-ruminant animals.
You did the partial of that work, so I don't know what the difference of your work is in essence? Moreover, are your results consistent with or contrary to theirs?
2. Same question again. They studied Keratin genes and you did too. So where are your innovative points?
3. You used pig GIT gene expression data, but you did not note the source of this data in the Methods. Please add it in your text.
4. Did authors check and compare sheep results to bovine?
5. For reference format, why do some have doi and some not? Please unify reference according to the standard requirements of the Journal.
6. There are numerous typographical/grammatical errors in the manuscript, which should be corrected before publication. eg. Is an interrogative sentence at line 311?

Reviewer 2 ·

Basic reporting

•The manuscript is clear and written in a professional English.
•The introduction section includes sufficient information to show the context of the study.
•In general, the structure of the manuscript conforms PeerJ standards although for the in-text citations the authors should include a comma between the name and the year.
•Figures are high quality, well labeled and described. In general, the figures in the manuscript are relevant and they help to understand the different results obtained in the study, although, in my opinion, Figure 5 could be dispensable.
•Raw data adheres the PeerJ policy, as no primary datasets were generated in this study (they were downloaded from public repositories) and the secondary datasets obtained from the analysis are included in supplementary material.

Experimental design

•The manuscript is an original primary research within the Scope of PeerJ.
•The research question is well defined, relevant and meaningful, trying to elucidate the developmental origin of the rumen, the distinctive features of ruminant gastrointestinal tract (GIT) as well as the common features with non-ruminant species (pig and human).
•The investigation has been performed rigorously. Methods are described in detail although some comments regarding methods section are detailed above.
•An important aspect to be clarified is the age of the lamb used (is a lactating or adult lamb?

Validity of the findings

•Data on which conclusions are based is provided in public repositories and conclusions are well stated, linked to original research question and limited to supporting results. However, to assess the validity of the findings the authors should clarify several points regarding to the primary dataset which are detailed in comments section.

Additional comments

Major comments:
1. Primary Dataset:
a. Authors should specify the number of RNA-seq replicates available for each of the analyzed tissues. It is indicated that 26 tissue samples are included, but it is not specified which tissues have the three biological replicates (ram, ewe and lamb) and which not.
b. The authors indicate that tissue samples were obtained from trio of Texel sheep (ram, ewe and their lamb). I guess if lamb samples should be included in the analysis of GIT, dietary conditions and age of the lamb included in the analysis should be known.
2. Statistical analysis. It is indicated that raw count data was normalized and clustered DESeq2 and/or EdgeR. I understand that as the PCA analysis is done with DESeq2 the data is normalized with DESeq2 and for the ANOVA analysis done with EdgeR data is normalized with EdgeR but the “and/or” expression sound ambiguous. Furthermore, did the authors try to perform the differentially expression analysis with DESeq2? Are the results similar?
3. Length of the paper: Results and Discussion section is too long and should be drastically shortened and focused mainly in the paragraphs describing genes highlighted in figures 3 and 4.
Minor comments:
1. Line 95. Gastrointestinal tract has been already defined, use GIT.
2. Data acquisition and statistical analysis. I recommend to start this sub-section indicating that no primary datasets were generated in this work, to clarify the whole paragraph.
3. Co-expression network analysis. Authors should specify which transcripts are using for this analysis, the totality of transcripts or the significant genes obtained from the previous statistical analysis.
4. Line 167. FPKM definition: Fragments Per Kilobase Of Exon Per Million Fragments Mapped
5. Epidermal Development Complex (EDC) term should be defined in line 225 not in 228
6. GO enrichment analysis: Table 1 and Table S3. The first term that is indicated in Table 1 is EDC and when you look at table S3 genes related to EDC are not there. It is clarified in the text that no significant enrichment of genes in the EDC was identified in the GO analysis and I understand that these genes were not in the Table S3, but I miss some additional table with the genes related to EDC.
7. Line 257. Change Kallikrien to Kallikrein
8. Comparison of the sheep and pig GIT transcriptome. This sub-section sounds interesting but I feel that it is in the middle of the sub-sections describing genes involved in innate immunity and epithelial development genes (descriptions of genes represented in Figure 3) and the paragraphs describing genes involved in metabolic processes involved in GIT (descriptions of genes represented in Figure 4). I would change it to the end or after the sub-section titled “Identification of common and specific GIT and epithelial transciptomic signatures”
9. Host Micro-organism interactions. This sub-section describes genes which expression is represented in Figure 3, I would move this subsection to put it after “The stratified squamous rumen epithelium expression signature” in order to group the two sub-sections that describe genes represented in Figure 3.

·

Basic reporting

No comments.

Experimental design

No comments.

Validity of the findings

No comments.

Additional comments

Review on #8247: Epithelial, metabolic and innate immunity transcriptomic signatures differentiating the rumen from other sheep and mammalian gastrointestinal tract tissues, by Xiang et al.
The authors analysed the transcriptomes of rumen, abomasum, duodenum, caecum, colon and rectum, two other stratified squamous epithelia, skin and tonsil, non-epithelial tissue, liver and muscle and spleen representing immunological active tissue. These transcriptomes were compared with data obtained in pigs and humans. The aims were to identify characteristic features of the rumen, shared features and the developmental origin of the rumen. The manuscript is of importance for a broad range of researchers; however, parts of it are difficult to read for those who are not familiar with genetics (like me, being a physiologist…).
He authors give the key issue (26-28): “How has the presence of the rumen affected other sections of the gastrointestinal tract of ruminants compared to non-ruminants?”
This is an interesting question, but I am sometimes missing the link between genetics and physiological function. If you want to work out the “ruminant-specific” aspects, why don´t you compare ruminants to other herbivores? Or pre-ruminating lambs to adult animals? I understand that the pig was chosen because of its close relationship, but what about humans? Wouldn´t it be interesting to compare the sheep to the horse instead?
There is some inaccurateness in respect to ruminant physiology:
71-71: Lipids are not completely converted into SCFAs.
77-78: As far as I know, nucleases are secreted by the pancreas, not in the abomasum.
80-82: Depends on the ration: corn starch is very different from barley.
113: Which part of the rumen has been used for analysis?
257: kallikrein
273-274: Progesterone is a steroid.
275-276: Progesterone exerts its effects via a nuclear receptor. How would it influence the ruminal microbiota? Might it be that the rumen contributes to steroid elimination? The expression of members of the UGT family might also point to this idea, the rumen as an organ involved in the elimination of different compounds, like the liver.
291: Comparison of the sheep and pig transcriptomes: Were there any differences related to the more constant flow of ingesta in ruminants?
302-306: Were there any interesting differences between sheep and pigs? I would assume that gene products related to intestinal glucose transport should differ.
311-387: I like this part of the results & discussion section.
389-390: In fact, a ration adequate for ruminants contains less than 5% crude fat – otherwise you would get a negative effect on ruminal microbiota.
420-422: There is a lot of discussion about the capacity of the small intestine to adapt to large amounts of by-pass starch. In this context, your findings in respect to carbohydrate digestion are very interesting.
443-450: Does this mean that there is a relevant absorption of glucose in the large intestine? In line 451-454 you state that the expression pattern of SGLT1 is comparable to that of monogastric animals.
464-465: Did you take into account that urea is also secreted via saliva?
514: stomach: abomasum?
557-561: Could you please explain the function of PIP?

---

## Round 0.2 · accepted · Accept

· Academic Editor

Accept

I have only a few minor edits for you to perform (line numbers refer to the document 'peerj-8247-revised_manuscript_text_Xiang_et_al_11022016_make-up.docx):

l. 137: change 'information were specified' to 'information are specified'

l. 150: change 'was downloaded' to 'were downloaded'

l. 165: change 'analysis' to 'analyze'

l. 168: Period after the linear model should be removed, and 'Where' should be lower case